# Polyhydroxybutyrate (PHB) Scaffolds for Peripheral Nerve Regeneration: A Systematic Review of Animal Models

**DOI:** 10.3390/biology11050706

**Published:** 2022-05-05

**Authors:** Maria Florencia Lezcano, Giannina Álvarez, Priscila Chuhuaicura, Karina Godoy, Josefa Alarcón, Francisca Acevedo, Iván Gareis, Fernando José Dias

**Affiliations:** 1Research Centre in Dental Sciences (CICO-UFRO), Dental School—Facultad de Odontología, Universidad de La Frontera, Temuco 4780000, Chile; florencia.lezcano@ufrontera.cl (M.F.L.); giannina.alvarez@ufrontera.cl (G.Á.); josefa.alarcon@ufrontera.cl (J.A.); 2Department of Integral Adults Dentistry, Dental School—Facultad de Odontología, Universidad de La Frontera, Temuco 4780000, Chile; priscila.chuhuaicura@ufrontera.cl; 3Departamento de Bioingeniería, Facultad de Ingeniería, Universidad Nacional de Entre Ríos, Oro Verde 3100, Argentina; igareis@ingenieria.uner.edu.ar; 4Oral Biology Research Centre (CIBO-UFRO), Dental School—Facultad de Odontología, Universidad de La Frontera, Temuco 4780000, Chile; 5Scientific and Technological Bioresource Nucleus (BIOREN-UFRO), Universidad de La Frontera, Temuco 4780000, Chile; karina.godoy@ufrontera.cl; 6Department of Basic Sciences, Faculty of Medicine, Universidad de La Frontera, Temuco 4780000, Chile; francisca.acevedo@ufrontera.cl; 7Center of Excellence in Translational Medicine (CEMT), Faculty of Medicine, Scientific and Technological Bioresource Nucleus (BIOREN), Universidad de La Frontera, Temuco 4780000, Chile

**Keywords:** neural repair, biomaterial, PHA—polyhydroxyalkanoate, peripheral nervous system—PNS

## Abstract

**Simple Summary:**

Currently, polymeric biomaterials are the choice for the design of scaffolds for the regeneration of peripheral nerves. Polyhydroxybutyrate (PHB) is a polymer belonging to the class of polyesters that are produced naturally in nature by microorganisms. To gain a better understanding of the efficacy of therapeutic approaches involving PHB scaffolds for peripheral nerve regeneration, we conducted a systematic review of the literature with the aim of discussing the current knowledge of PHB scaffolds applied to nerve regeneration. The use of PHB as a biomaterial to prepare tubular scaffolds for nerve regeneration was shown to be promising. The incorporation of additives appears to be a trend that improves nerve regeneration.

**Abstract:**

In the last two decades, artificial scaffolds for nerve regeneration have been produced using a variety of polymers. Polyhydroxybutyrate (PHB) is a natural polyester that can be easily processed and offer several advantages; hence, the purpose of this review is to provide a better understanding of the efficacy of therapeutic approaches involving PHB scaffolds in promoting peripheral nerve regeneration following nerve dissection in animal models. A systematic literature review was performed following the “Preferred Reporting Items for Systematic Reviews and Meta-Analyses” (PRISMA) criteria. The revised databases were: Pub-Med/MEDLINE, Web of Science, Science Direct, EMBASE, and SCOPUS. Sixteen studies were included in this review. Different animal models and nerves were studied. Extension of nerve gaps reconnected by PHB scaffolds and the time periods of analysis were varied. The additives included in the scaffolds, if any, were growth factors, neurotrophins, other biopolymers, and neural progenitor cells. The analysis of the quality of the studies revealed good quality in general, with some aspects that could be improved. The analysis of the risk of bias revealed several weaknesses in all studies. The use of PHB as a biomaterial to prepare tubular scaffolds for nerve regeneration was shown to be promising. The incorporation of additives appears to be a trend that improves nerve regeneration. One of the main weaknesses of the reviewed articles was the lack of standardized experimentation on animals. It is recommended to follow the currently available guidelines to improve the design, avoid the risk of bias, maximize the quality of studies, and enhance translationality.

## 1. Introduction

Peripheral nerve injury is a problem of high incidence that implies loss of both sensibility and motor function [1,2]. When the injury causes neurotmesis—i.e., discontinuity of the nerve—surgical techniques must be applied in an attempt to repair the damage [3]. The most commonly used techniques consist of directly suturing the ends of the cut nerve or closing the gap with an autograft—i.e., placing a piece of nerve harvested from the same individual—or an allograft—i.e., a piece of nerve harvested from another individual of the same species [2,3]. The ends of the nerve can only be sutured when the gap is short [2,3]. The use of autografts or allografts to close wider gaps has the disadvantage of causing denervation or requiring a donor [2,3]. An interesting alternative to close extensive gaps is the placement of an artificial scaffold with the characteristics of promoting nerve regeneration [2].

Scaffolds are biomaterials designed to perform the following functions during tissue regeneration: (i) promote cell-biomaterial interaction, cell adhesion, and extracellular matrix deposition, (ii) allow the passage of gases, nutrients, and regulatory factors to allow cell survival, proliferation, and differentiation, (iii) biodegrade to a controllable rate similar to the rate of tissue regeneration under the conditions of interest, and (iv) cause minimal toxicity in vivo [4]. The design of the scaffold is very important since it determines its properties, causing it to be more efficient in specific applications [5]. One of the major challenges in developing a scaffold lies primarily in the choice of a biomaterial with suitable properties [6].

Different promising alternatives for building scaffolds have been described in the literature, including peptides [7], glycoderivatives [8], and synthetic materials such as polyrotaxanes [9]. Polymeric biomaterials are widely preferred when it comes to scaffold design for peripheral nerve regeneration [6]. In the last two decades, artificial scaffolds for nerve regeneration have been produced using a variety of polymers, both natural and synthetic, and biodegradable and non-biodegradable [10]. Among the biodegradable and biocompatible polymers, polyhydroxyalkanoates (PHAs; Figure 1) stand out as they are a family of polyhydroxyesters of 3-, 4-, 5- and 6-hydroxyalkanoic acids, which are produced by a variety of bacterial species under nutrient-limiting conditions with carbon excess and are found as discrete cytoplasmic inclusions in bacterial cells. For this reason, they are considered green plastics and have a positive social and environmental impact [11,12].

Poly-3-hydroxybutyrate (P3HB) is a very promising biodegradable polymer of the polyhydroxyalkanoate (PHA) family for making nerve scaffolds [13,14,15,16,17,18,19,20,21,22,23,24,25,26,27,28,29,30,31,32,33,34]. The advantages of P3HB over other polymeric materials that give it many possibilities of being an ideal material include their synthesis from renewable sources, good tensile strength, good flexibility, and excellent biocompatibility and biodegradability properties, properties necessary in biomedical applications [29,33,35,36]. It is also a versatile natural polymer that can be extruded, molded, spun into fibers, made into films, and mixed with other polymers to produce heteropolymers. This has made the selection of the P3HB biopolymer a natural choice to create 3D structures suitable for fabricating tissue engineering scaffolds [12].

**Figure 1 biology-11-00706-f001:**
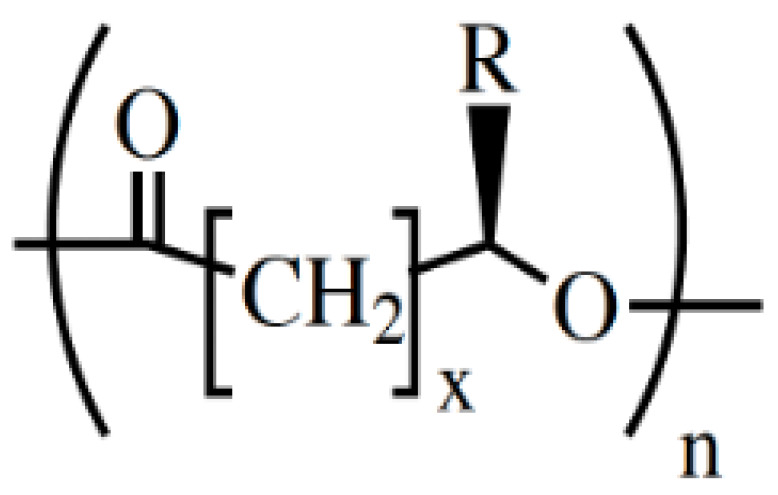
General chemical structure of the PHAs. Typical values: x = 1 to 4; *n* = 1000 to 10,000; R = alkyl group (C_m_H_2m+1_) or functionalized alkyl group. Reused from [37] with kind permission of John Wiley and Sons.

PHB in its simpler form, i.e., Poly(3HB) (Figure 2), became widely available at the beginning of the 1990s, which provided opportunities for its evaluation as a medical biomaterial [37]. At the beginning, it was not targeted as an implantable biomaterial and was thus lacking the quality for obtaining approval of drug administrators [38]. PHB causes prolonged and acute inflammatory responses, so the need was to produce PHB of high purity, check its biodegradation in vivo, conduct the fabrication of scaffolds and modify their surface [38]. The first film made of PHB for surgical applications was approved by the FDA in 2007 [39].

The advantages of using PH3B are inherent; however, one of the main challenges faced by researchers engineering tissues from natural polymers is the design of scaffolds with desirable physical and mechanical properties for the growth and proliferation of cells [40]. P3HB represents a great industrial and scientific advance in the search for new sustainable energy sources.

Based on this premise, this systematic review will discuss the current knowledge of PHB scaffolds applied to nerve regeneration. The focus is to provide a better understanding of the efficacy of therapeutic approaches involving PHB scaffolds in promoting peripheral nerve regeneration following nerve dissection in animal models.

## 2. Materials and Methods

### 2.1. Literature Search Strategy

A systematic literature review was performed following the “Preferred Reporting Items for Systematic Reviews and Meta-Analyses” (PRISMA) criteria [34]. The revised databases were: PubMed/MEDLINE, Web of Science, Science Direct, EMBASE, and SCOPUS. The following query was used: (“PHB” OR “P3HB”) AND (“nerve injury” OR “nerve lesion” OR “nerve regeneration”). A manual search of the literature was carried out by reviewing the references in the articles found in the electronic search. The search was performed between September 2019 and April 2022.

### 2.2. Eligibility Criteria

This systematic review includes primary in vivo studies analyzing the use of PHB scaffolds applied to achieve peripheral nerve regeneration in lesions of the type of nerve resection. No publication date limit was selected. Studies in English conducted on animals were considered, as they were the most numerous in this topic and because they can be analyzed regarding their quality associated with methodology/results and risk of bias using available instruments (ARRIVE guidelines [41] and SYRCLE RoB [42] respectively).

### 2.3. Article Selection

The articles obtained in the systematic search were analyzed by reviewing the titles and abstracts. Those articles that met the eligibility criteria were analyzed in full text to confirm their relevance to the topic analyzed.

### 2.4. Data Extraction

Data were collected from full-text studies, in which relevant aspects of nerve injury and PHB nerve scaffolds were reported. The information collected was: Authors and year of publication; nerve (type)/gap size/time periods; animal model and sample size (n); PHB used; scaffold fabrication method; additives used (if any); methods used; conclusion or main outcome.

### 2.5. Analysis of Methodology and Results of Selected Studies

The analysis of the methodology and the results of the selected studies was performed by analyzing the full-text studies to determine whether some aspects of the ARRIVE (Animal Research: Reporting of In Vivo Experiments) guidelines were covered. The aspects considered were the ones relevant to the main focus of this review: Ethics, the authors declare following ethical guidelines for animal experimentation or having approval by a Scientific Ethical Committee; control, experimental protocol has a control group; control 2: experimental protocol has 2 control groups; PHB type: the authors declare the type of PHB used (e.g., Polyhydroxybutyrate, Poly-3-hydroxybutyrate, etc.); PHB origin: the authors declare the origin of the PHB used; scaffold fabrication method: the authors explain the method of scaffold fabrication; nerve gap size: the authors declare the size of the nervous gap; nerve studied: the authors declare which nerve was studied; time periods: the authors declare in which time periods were the evaluations carried out; surgical procedure: the authors explain the surgical procedures performed; euthanasia method: the authors explain the euthanasia method used; species: the authors declare the animal species studied; sex/weight: the authors declare sex and weight of animals at the beginning of the experimental protocol; group size and distribution: the authors declare the size of the experimental group and explain the distribution of animals in the groups; group size justification: the authors justify the sample size; statistics: the authors declare the statistical methods used; complete results: the authors present the complete results of the proposed methodology; precision measures: the author report the precision values of the quantitative data (e.g., SD, SEM, IQ distance); limitations: the authors state the limitations of the study; conclusion > objectives: the conclusion is consistent with the proposed objectives. The analysis using the ARRIVE guidelines was presented in the form of a table in order to show if the items analyzed were covered by each individual study or not. In the end, a score of the percentage of compliance was calculated for each study and included in the results table.

### 2.6. Analysis of the Risk of Bias of Selected Studies

The analysis of the risk of bias of the selected studies was carried out in the full-text documents using the SYRCLE RoB (Systematic Review Center for Laboratory Animal Experimentation—Risk of Bias) tool, which was also adapted to the main focus of this review. The dimensions analyzed for each study were: selection bias; performance bias; detection bias; attrition bias; reporting bias and other bias sources. The results of the risk of bias of the selected studies using the SYRCLE RoB tool were presented in the form of a table showing the fulfilled (Y), unfulfilled (N), and uncertain (U) dimensions. The ARRIVE and SYRCLE RoB analyses were performed by two independent examiners. In cases where there was disagreement between the examiners, a third examiner was consulted.

## 3. Results

### Study Selection

The article search and selection process is summarized in Figure 3. The total number of articles found in the databases was 193 citations, of which 82 articles were duplicates. After the initial reading by title and abstract, 71 articles were ruled out for being unrelated to the review topic. After reading full-text articles (40 in total), 24 were excluded. Finally, in this review, 16 articles were included that corresponded to experimental studies in vivo that met the previously defined criteria.

Data were extracted from studies in selected animals using Table 1 which considers information relevant to nerve regeneration studies using tubular scaffolds prepared with PHB.

Studies using PHB as a biomaterial for the preparation of tubular scaffolds used in peripheral nerve regeneration have been developed since the 1990s [13,14,15,16]. Several animal models have been used, with most studies carried out on rodents and rabbits. Likewise, different nerves have been studied, with the sciatic nerve being the most studied of all. It is also noted that the sizes of nerve gaps reconnected by PHB tubular scaffolds and the periods of analysis were varied.

The most common PHB form used in the selected studies was “poly-3-hydroxybutyrate.” The most common way of manufacturing the scaffolds was rolling a purchased PHB sheet to form a tube (Figure 4 and Figure 5). The sheets were made by Astra Tech AB (Mölndal, Sweden), had a molecular weight of approximately 150 kD, and unidirectional fiber orientation. Only three studies manufactured the PHB scaffolds autonomously [13,25,34]. Regarding the additives included in the scaffolds, a trend was repeated in almost all studies from 2003 to 2018: the adding of growth factors, other biopolymers, and neural progenitor cells.

Most of the selected studies analyzed morphological characteristics using different methods—e.g., histology, immunohistochemistry, and ultrastructure. Functional aspects of nerve recovery were analyzed only in the two most recent studies [33,34].

The use of PHB as a fabrication material for tubular nerve scaffolds (NC, NGC) seems to be promising for peripheral nerve regeneration [13,16] in cases of long-gap nerve injury [17] generating a rapid reconnection of these nerve gaps [23]. The studies reported adequate properties of PHB in terms of mechanical strength, biodegradability, biocompatibility, and support for the proliferation and growth of neurons [25,27]. However, in some cases, no differences were noted with the standard surgical treatment of epineural repair (epineural suture) [14,15]. In other cases, other materials, such as fibrin, obtained better results [24]; even PHB without additives inhibited peripheral nerve regeneration [18]. The inclusion of additives such as GGF [18,21], rhLIF [19], and cells [22,33,34] in the PHB scaffolds suggest an improvement in the nerve regeneration results obtained.

The analysis of the quality of the studies through the application of the ARRIVE guidelines adapted for this review is summarized in Table 2. Some of the analyzed aspects showed a low level of compliance in the studies. For example, “sample size justification” was not met in any study; the inclusion of a “second control group” was present in three [25,27,34] of the sixteen studies; the limitations were declared in four [13,14,18,20]; euthanasia methods in six [14,18,20,33,34]; and the sex/weight of the animals in eight [13,15,18,20,21,25,27,34]. In addition, six studies [13,15,22,24,27] present a conclusion that disagrees with the objectives set and four [16,20,21,27] of the included studies did not inform of the statistical methods used.

The risk of bias analyzed by applying the SYRCLE RoB tool (Table 3) showed that the criteria of the dimensions “Performance” and “Detection” were “Unclear” in all parameters. In the “Selection” dimension, only the “Baseline” parameter was met in half of the studies. In practically all selected studies, it was not clear if the “Allocation” and “Sequence” parameters were covered. Four studies fulfilled the “Attrition” parameter because they declared that there were no animal losses and/or explained the losses that occurred [14,23,24,34]. Five studies did not clearly explain animal losses, but reported the postoperative characteristics of the experimental protocols [18,19,20,21,33]; and seven studies did not deliver clear information for animal losses occurred in the experiments [13,15,16,17,22,25,27]. In the “Reporting” dimension, only nine studies declared the positive and negative results of the experiments performed [13,14,18,20,21,23,27,33,34].

Finally, seven studies showed other possible sources of bias (Other/Free of Other Problems). Borkenhagen et al. did not state a clear objective or hypothesis of the study [13]. Ljungberg et al. showed a correlation that is not described in the methodology [14]. Hazari et al. presented data on controls selectively [15]. In addition, six studies showed a close relationship with the company that developed the PHB sheets used [14,15,16,19,21,22]. Furthermore, in two of them, an author was affiliated with the company [15,16]; in another two, the authors declared being financed by the company [14,21]; and finally, the PHB used was donated by the company in two other studies [19,22].

## 4. Discussion

The present systematic review gathered and analyzed evidence on the use of polyhydroxybutyrate biopolymer (PHB), a type of polyhydroxyalkanoate (PHA), in peripheral nerve regeneration. PHB is a biopolymer of microbial origin [43] that have several positive properties in their use as a material in the construction of nerve scaffolds, such as adequate flexibility and tensile strength, excellent biocompatibility, and biodegradability [29,33,35,36].

This systematic review focused mainly on PHB because it is the most used PHA for this type of application. After the initial literature search, several in vitro and animal studies and only one clinical study were found [26]. Thus, the authors decided to focus on animal studies due to the number of studies available and the existence of well-known tools to analyze quality (ARRIVE) and risk of bias (SYRCLE RoB) for these types of studies.

The in vitro studies that evaluated the use of PHB as a biomaterial for the construction of peripheral nerve scaffolds are of fundamental importance in this topic, mainly helping to understand cellular behaviors in obtaining the best regenerative results [29,44,45,46,47]. However, these studies show high heterogeneity in their methods and designs, which makes it difficult to compare their results. In recent studies, it was noted that Schwann cells, the glial cells supporting the peripheral nervous system, were the most studied cell type [44,45,46]. PC12 [29], neuronal [46], and fibroblast lineage [47] cells that produce the collagen that makes up the endo and perineurium region were also studied with the intention of understanding the interaction of PHB with peripheral nerve regeneration.

The method of fabrication of PHB scaffolds chosen by these studies was preferentially electrospinning [29,44,45,46,47]. The cell analysis strategies used were mainly viability [29], differentiation [44,46] proliferation [29,45,47], migration [47], adhesion [29], and cell growth orientation [44,45,46]. Better results were observed in the cases where PHB was associated with some approaches to improve the properties of this material, such as the mixture/blend of other polymers (PHBV [44,45], P(3HO), and P(3HB) [46]) and polyaniline [47], in addition to the functionalization of the PHB scaffolds using oxygen plasma printing [47] and naturally occurring peptides in the ECM [45].

The ARRIVE and SYRCLE RoB scores revealed a good quality of studies carried out in animals with the tubular PHB nerve scaffold. However, weaknesses were noted in these studies. The ARRIVE application revealed a weakness present in all selected studies: the lack of justification for the sample size, which is a factor that directly affects the statistical power associated with the study and, therefore, the reliability of the conclusions. In addition, only four studies declared their limitations [13,14,18,20], six studies declared the method of euthanasia applied [14,18,20,21,33,34] and eight studies [14,15,18,20,21,25,27,34] stated the weight and sex of the animals. This lack of information also influences the validity and mainly the reproducibility of the studies carried out. Finally, it was noted that six studies [13,15,20,22,24,27] presented a conclusion that did not answer clearly the objective of the study and four studies [16,20,21,27] did not state clearly the statistical methods used. This lack of information may raise doubts about the results of these studies.

The application of the SYRCLE RoB tool to assess the risk of bias in the selected studies revealed that many of the criteria and dimensions evaluated did not present sufficiently clear data or information. The parameter of “Baseline” were fulfilled by seven studies [14,15,18,20,21,27,34]. The “Reporting bias” dimension was met by nine studies [13,14,18,20,21,23,27,33,34] and “Attrition bias” was met in only four studies [14,23,24,34]. A fact that drew attention in the analysis of the risk of bias was the close relationship between authors and the company Astratech in six studies [14,15,16,19,21,22]. Astratech developed the PHB sheets used in the studies and provided funding. In all these cases, the results of using this biomaterial in nerve regeneration were positive or promising.

The application of the two tools (ARRIVE and SYRCLE) for the analysis of studies carried out in animals revealed the weaknesses of these studies in relation to their methodology, results, and risk of bias. This lack of success to comply with all parameters and dimensions analyzed by both tools does not necessarily completely invalidate the results of these studies. Most of the studies selected in this review were published prior to ARRIVE (2010) and SYRCLE (2014). However, that does not mean that they were exempted from complying with the conditions needed to reach reliable results. There must be future efforts to comply with the parameters of tools like ARRIVE and SYRCLE in order to favor the standardization, quality, and, therefore, relevance of studies in animals. The studies in animals have evolved to favor their translationality to clinical studies in the last years [48,49,50]. The data extracted from the selected studies showed that the use of PHB as a biomaterial for preparing nervous scaffolds dates back to the 1990s [13,14,15,16]. Most of the reviewed articles were published between 1995 and 2010. Then, the number of articles started to drop and only two of them were published in the last 10 years.

The model of nerve injury studied in all cases is that of nerve resection, in which one gap is generated in the nerve, which can be considered the most serious type of nerve injury because it generates a total nerve interruption (neurotmesis) with the problem of lack of biological structure for its reconnection. However, it was noted that the studied nerve, the gap size, and the recovery times were varied among studies. In most cases, the studied nerve was classified as “mixed”—e.g., sciatic, radial, peroneal, recurrent laryngeal—which favors the analysis of both motor and sensory recovery.

The size of the gap ranged from 2–3 mm in cats [14,15] to 40 mm in rabbits [17,18], with the 10 mm gap in rats being the most evaluated model [16,19,22,23,24,25,27,33,34]. This could indicate a possible standardization of the gap among the most recent studies carried out in rats. The post-nerve injury analysis times were also very varied, from 7–30 days [16] to 6–12 months [14,15] without an apparent pattern or trend in the selected studies. Previous studies have shown that the larger the size of the nerve gap, the worse the prognosis of nerve regeneration [51].

Regarding the manufacture of the scaffolds, 13 studies used commercial PHB sheets made by electrospinning [14,15,16,17,18,19,20,21,22,23,24,27,33]. Only three studies manufactured their own custom-made scaffolds for nerve regeneration by the methods of melt extrusion [13], dipping-leaching [25], and electrospinning [34]. The own manufacture of the scaffolds may not be as standardized as the use of PHB sheets industrially assembled by electrospinning; however, it would allow the customization of the characteristics of the scaffolds in order to improve regeneration results, especially with the intention of preparing an environment favorable to nerve cells because it resembles the extracellular matrix. The preparation of scaffolds by the electrospinning method allows obtaining a structure similar to the extracellular matrix, which would help to promote nerve regeneration [52]. Most of the studies did not discuss critical characteristics of the material such as molecular mass and purity. Further characterization of the material used is needed in order to understand its impact on the treatment success. PHB can cause prolonged and acute inflammatory responses, so the need is to produce PHB of high purity, check its biodegradation in vivo, improve the fabrication of scaffolds and modify their surface [38].

Most of the included studies analyzed improvement of nerve regeneration by morphological methodologies exclusively, such as macroscopic morphology, histology, microscopic morphometry, immunohistochemistry, and ultrastructural analysis [3,4,5,6,7,8,9,10,11,12,13,14,15,16,17,18,19,20,21,22,23,24,27]. These methods are known for their application in studies of changes in peripheral nerves, allowing some prediction of functional characteristics [53,54]. The main objective of using PHB scaffolds would be the restoration of the function lost by peripheral nerve damage, which remains one of the main challenges in nerve regeneration [55,56,57,58]. Studies analyzing sensory and motor function by direct methods are necessary to assess the effectiveness of nerve regeneration achieved by the use of this type of biomaterial. However, this was done only in two of the most recent studies [33,34].

The main results of the selected studies revealed in most studies that the use of PHB for peripheral nerve regeneration was positive and or promising, with the adequate features of mechanical strength, biocompatibility and proliferation of neurons [13,16,17,18,19,20,21,22,23,25,27,33,34]. Studies carried out in rodents and those that included additives showed practically only positive or promising results from the use of PHB as a biomaterial for nerve regeneration. The inclusion of additives seems to be a trend that has been observed in most of the more recent studies [18,19,21,22,23,27,33,34], which seems to reveal that PHB nerve scaffolds would serve more as a vehicle for other nerve regeneration strategies, such as the adding of growth factors and cells.

Within the selected studies, the lack of control groups in five studies drew attention [13,18,21,23,24]. The use of control groups was proposed in 1801 and is currently standard practice in basic and clinical studies [59]. The control group consists of a group with the same configuration as the experimental groups except for the applied/studied variable [60]. The lack of control groups makes the quality of these studies questionable, as a good control group validates the experiment carried out and provides the basis for evaluating the effects of treatments [61] that are necessary when studying peripheral nerve regeneration using scaffolds. In addition, in these cases, two simultaneous control groups could be used, the negative control groups—i.e., lack of intervention—and positive control groups—i.e., intervention with expected effect [61]—in order to establish the effect of using PHB scaffolds in a more solid and objective way. However, among the selected studies only three recent studies used two control groups in their methodology [25,27,34].

Only one clinical study was found on this topic, Åberg, et al. [26], which compared the use of PHB tubular scaffolds with the epineural suture in 12 individuals through sensory and motor clinical, neuropsychological, morphological, and psychological tests after 2 weeks, 3, 6, 9, 12 and 18 months after surgery. The study revealed that most tests showed no differences between treatments, but suggested an improvement in sensory recovery in patients who received the PHB scaffold, although it would need confirmation in a larger clinical study. This result agrees in part with the animal studies analyzed in the present review; in addition, it is worth mentioning that the scaffold used in this clinical study performed by Åberg, et al. [26] was not associated with any additive, which could change its results [26].

Among the limitations of this systematic review, we can mention searching for studies published only in English, not reviewing published or unpublished studies such as theses and dissertations, or even in different databases from those consulted. In addition, this review was focused only on animal studies, which was the authors’ decision as they found only one clinical study on this topic.

## 5. Conclusions

The use of PHB as a biomaterial to prepare tubular scaffolds for nerve regeneration was shown to be promising. The incorporation of additives appears to be a trend that improves nerve regeneration.

One of the main weaknesses of the reviewed articles was the lack of standardized experimentation on animals. It is recommended to follow current guidelines, such as ARRIVE and SYRCLE RoB, to improve the design, avoid the risk of bias, maximize the quality of studies, and enhance translationality. In particular, control groups should be defined more carefully.

The experimental methodologies found in the literature were shown to be highly heterogeneous. To achieve more conclusive results, it would be recommended to standardize the injuries and the time periods analyzed in future studies. It would also be recommended to conduct functional analysis in order to directly assess the effectiveness of the treatments regarding function recovery.

Further efforts must be made in order to refine the fabrication of PHB mats intended for medical use. Chemical characteristics of the material should be addressed in future in vitro and in vivo studies in order to correlate them with the results. Critical aspects such as purity should be taken into account. Further characterization of the material used is needed in order to understand its impact on treatment success. Clinical studies should be conducted after proven success of the material in both in vitro and in vivo studies in animals.

## Figures and Tables

**Figure 2 biology-11-00706-f002:**
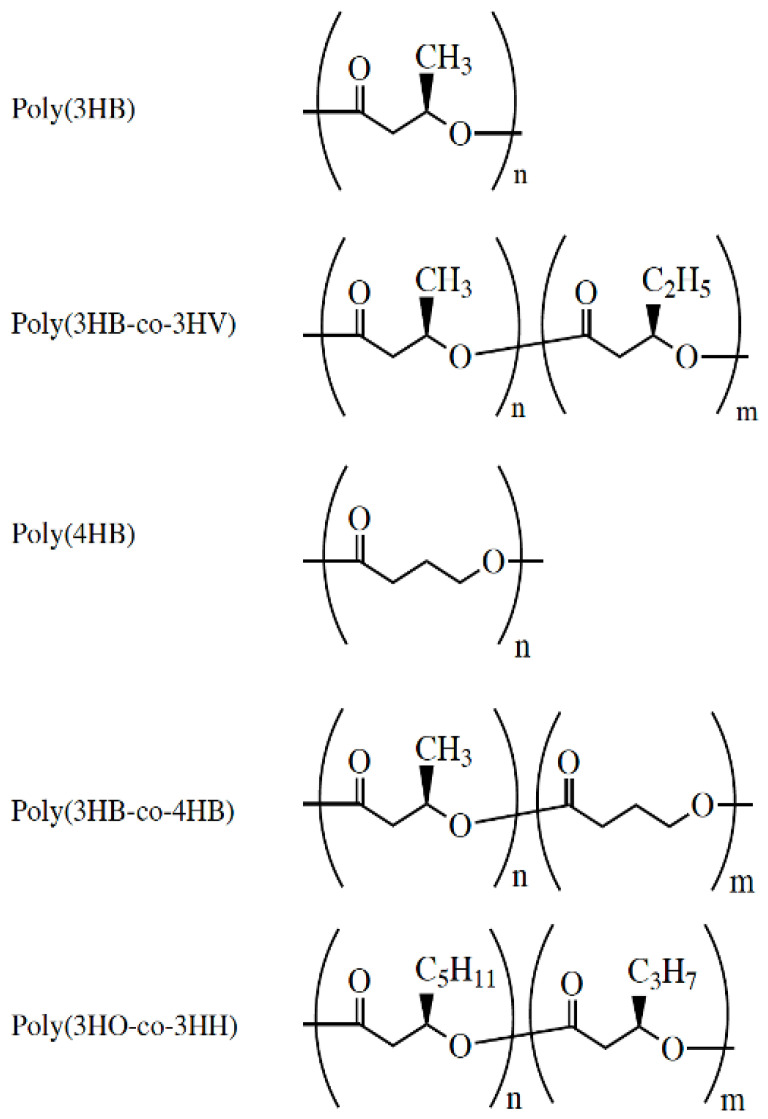
Chemical structure of PHAs under medical investigation. Reused from [37] with kind permission of John Wiley and Sons.

**Figure 3 biology-11-00706-f003:**
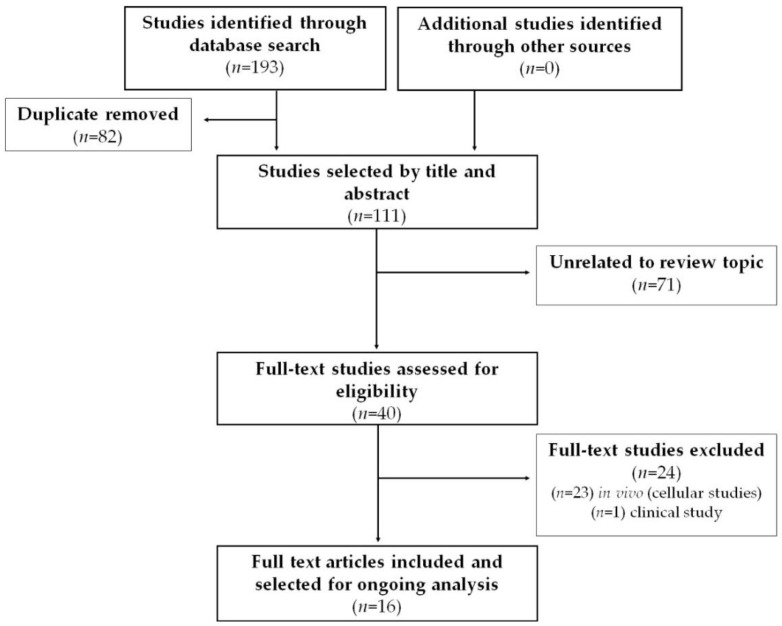
Systematic search flow diagram.

**Figure 4 biology-11-00706-f004:**
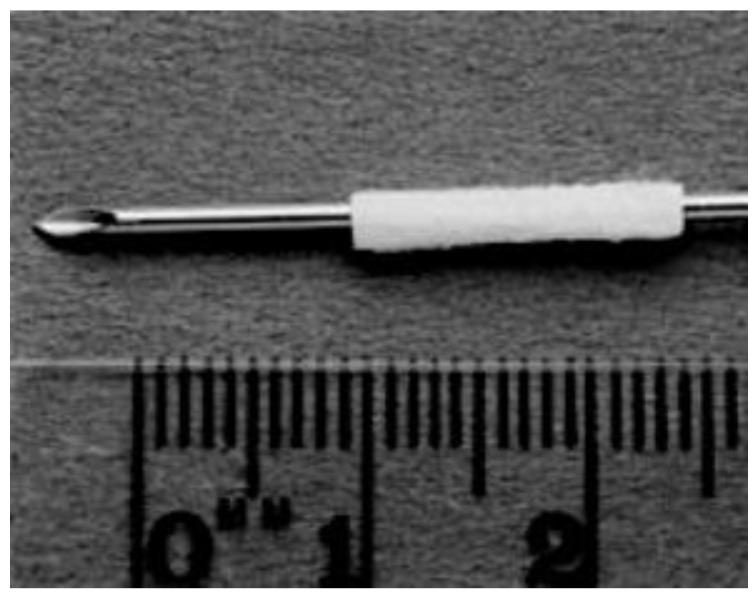
Unidirectional fiber orientated sheet of PHB (8 × 14 mm) rolled around a 16 G cannula to form a tube 14 mm long and with an internal diameter of 1.6 mm. Reused from (Hazari et al., 1999b) [16] with kind permission of Elsevier.

**Figure 5 biology-11-00706-f005:**
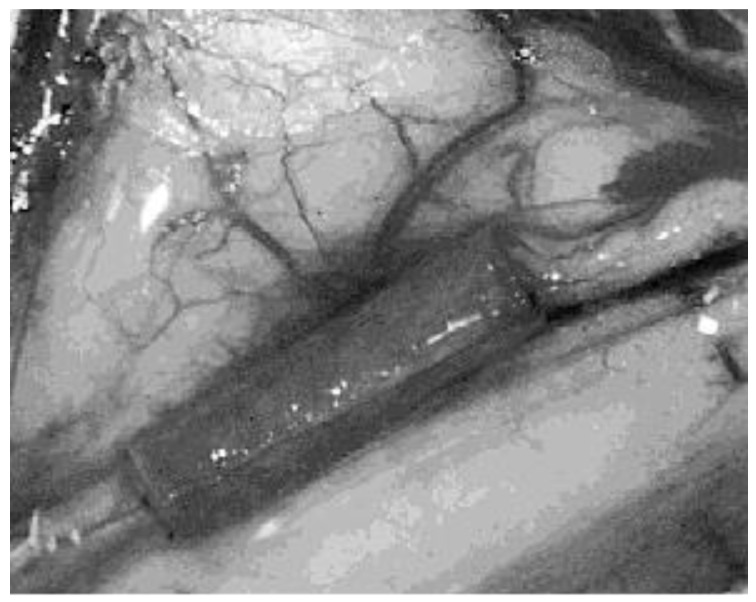
Superficial radial nerve of a cat sectioned and wrapped in a PHB conduit sealed by fibrin glue. Magnification 6×. Reused from (Ljungberg et al., 1999) [14] with kind permission of John Wiley and Sons.

**Table 1 biology-11-00706-t001:** Evidence extracted from animal studies selected.

Authors and Year of Publication	Nerve (Type)/Gap Size/Time Periods	Animal (*n*)	PHB Formula Used	Scaffold Fabrication Method	Additive (If Any)	Methods Used	Conclusion or Main Outcome
Borkenhagen et al., 1998 [13]	Sciatic nerve (mixed)/8 mm/4, 12, 24 weeks	Rats (26)	Poly[glycolide-co-(ε-caprolactone)]-diol & poly[(R)-3-hydroxybutyric acid-co-(R)-3-hydroxyvaleric acid]-diol (all polymers had a molecular weight higher than 100 KDa)	Melt extrusion (tube)	No additive	Macroscopic morphology, histology	PHB holds promises for its utilization as nerve guidance channels.
Ljungberg et al., 1999 [14]	Superficial radial nerve (sensory)/~2–3 mm/6, 12 months	Cats (20)	Polyhydroxybutyrate (PHB) (molecular weight 150 KDa)	Rolled sheets (tube) and fibrin glue.	No additive	Histology, quantitative immunohistochem. (IHC)	No differences between wrapping the nerve ends in PHB sheet or epineurally suturing of the nerve.
Hazari et al., 1999a [15]	Radial Nerve (mixed)/2–3 mm/6, 12 months	Cats (20)	Poly-3-hydroxbutyrate (PHB) (molecular weight 150 KDa)	Rolled PHB sheet wrapped around the nerve ends & Tissue Glue	No additive	Histology, quantitative IHC	No differences beetwen PHB tube and Epineural Repair
Hazari et al., 1999b [16]	Sciatic nerve (mixed)/10 mm/7, 14, 30 days	Rats (36)	Poly-3-hydroxybutyrate (molecular weight 150 KDa)	Rolled sheets sealed longitudinallywith cyanoacrylate (tube)	No additive	Quantitative IHC, morphometry	Good nerve regeneration in comparison with nerve grafts.
Young et al., 2002 [17]	Common peroneal nerve (Mixed)/2, 3, 4 cm/2, 3, 6, 9 weeks	Rabbit (90)	Poly-3-hydroxybutyrate (PHB) (molecular weight 150 KDa)	Sterile PHB sheets with unidirectional fiber orientation (long axes)	No additive	IHC, histology, macroscopic morphology	PHB conduits support peripheral nerve regeneration up to 63 days. They are suitable for long-gap nerve injury repair.
Mohanna et al., 2003 [18]	Common peroneal nerve (Mixed)/2, 4 cm/3, 6, 9 weeks	Rabbit (90)	Poly 3-hydroxybutyrate (PHB) (molecular weight 150 KDa)	Rolled PHB sheet around (16 G) cannula, long axes fiber orientation	Glial growth factor (rhGGF2, 1.29 mg mL^−1^, 80 kDa) diluted in 1 mL of 50:50 alginate fibronectin solution	Quantitative IHC	Inhibition of regeneration of nerve regeneration was partially reversed by the addition of GGF to the PHB conduits. PHB-GGF stimulates a progressive and sustainable regeneration increase in long nerve gap conduits.
Hart et al., 2003 [19]	Sciatic nerve (mixed)/10 mm/2, 4 months	Rats (30)	Poly-3-hydroxybutyrate (PHB) (molecular weight 150 KDa)	Rolled sheets (tube) PHB sheets—compressed PHB fibers (2–20 µm Ø)	Leukemia inhibitory factor (recombinant murine rhLIF 100 ng/mL) hosted in a matrix of hydrogel comprising 2% ultra-pure low-molecular-weight high-mannuronic-acid-content calcium alginate and 0.05% bovine fibronectin	Quantitative IHC, macro morphometry	rhLIF has a potential role in promoting peripheral nerve regeneration after secondary repair and can be effectively delivered within PHB conduits for nerve repair.
Birchall et al., 2004 [20]	Recurrent laryngeal nerve (mixed)/4 mm/30, 60, 120 days	Minipig (6)	Polyhydroxybutyrate (PHB) (molecular weight 150 KDa)	PHB sheet rolled to form a conduit	No additive	IHC; morphometry; histologic quantif.; macroscopic morphology	Functional and histological recovery within 2–4 months and appears to sustain abductor muscle fiber morphology. Recovery occurs despite a complex inflammatory response.
Mohanna et al., 2005 [21]	Peroneal (mixed)/20, 40 mm/120 days	Rabbit (30)	Poly-3-hydroxybutyrate (PHB) (molecular weight 150 KDa)	Rolled sheets (tube) PHB sheets—compressed PHB fibers (2–20 µm Ø)	Glial growth factor (rthGGF2, 1.29 mg mL^−1^, 80 kDa) diluted in 1 mL of 50:50 alginate fibronectin solution	Histology, quant. IHC, ultrastructure (TEM), muscle atrophy	GGF-containing PHB conduits promoted sustained axonal regeneration and improved target muscle reinnervation.
Kalbermatten et al., 2008a [22]	Sciatic (mixed)/10 mm/2 weeks	Rats (24)	Poly-3-hydroxybutyrate (PHB) (molecular weight 150 KDa)	PHB sheets rolled (16 G) 14 mm long, 2 mm diameter	A fibrinogen-cell solution was made in 1:10 dilution from Tisseel^®^ containing 9 mg/mL fibrinogen and 80 × 10^6^ Schwann cells/mL. This solution (25 mL) was used to coat PHB that was treated with 25 mL of diluted thrombin solution (5 IU/mL) for 10 min.	Histology, IHC, macroscopic morphology	Beneficial combinatory effect of an optimized matrix, cells and conduit material (PHB) as a step towards bridging nerve gaps.
Kalbermatten et al., 2008b [23]	Sciatic (mixed)/10 mm/2 weeks	Rats (12)	Poly-3-hydroxybutyrate (PHB) (molecular weight 150 KDa)	Rolled sheets of compressed PHB fibers soaked in fibrin glue (tube)	About 80 × 10^6^ Schwann cells/mL were suspended in 25 mL of fibrinogen solution. PHB conduits were coated with 25 mL of a diluted thrombin (5 IU/mL) solution for 10 min and then the fibrinogen/cell solution was added.	Histology, IHC, macroscopic morphology	PHB showed significant advantage in rapidly connecting a nerve gap lesion.
Kalbermatten et al., 2008c [24]	Sciatic (mixed)/10 mm/2, 4 weeks	Rodents (12)	Poly-3-hydroxybutyrate (PHB) (molecular weight 150 KDa)	PHB sheets wrapped around a cannula and heat sealed (tube) vs. Fibrin conduits.	No additive	Quantitative IHC, morphology	Advantage of the new fibrin conduit for the important initial phase of peripheral nerve regeneration in comparison with PHB conduit.
Bian et al., 2009 [25]	Sciatic (mixed)/10 mm/1, 2, 3 months	Rats (60)	Poly(3-hydroxybutyrate-co-3-hydroxyhexanoate) (PHBHHx) (molecular weight 610 KDa)	Dipping–leaching	No additive	Electrophysiol. analysis, histology, ultrastructure (TEM)	PHBHHx nerve conduits showed proper mechanical strengths and biodegradability artificial nerve conduits to repair nerve damages.
Durgam et al., 2010 [27]	Sciatic (mixed)/10 mm/8 weeks	Rats (11)	Poly(3-hydroxybutyrate-co-3-hydroxyvalerate) (PHB-HV) (no information on molecular weight)	Rolled sheets of PCL and PECA glued with a PHB-HV solution (tube)	Co-polymers of polypyrrole (PPy) with poly (ε-caprolactone) (PCL) and poly (ethyl cyanoacrylate) (PECA). Melt-pressed PHB-HV films were airbrushed with a PPy co-polymer (PPy–PCL or PPy–PECA) and pressed.	Histology	Biomaterials (PCL, PECA and PHB-HV) have good biocompatibility and support proliferation and growth neurons in vivo (without electrical stimulation).
Schaakxs et al., 2017 [33]	Sciatic (mixed)/10 mm/12 weeks	Rats (15)	Poly-3-hydroxybutyrate (PHB) (molecular weight 150 KDa)	Rolled sheets of compressed PHB fibers soaked in fibrin glue (tube)	Primary Schwann cells (SCs) isolated from adult rat sciatic nerves or SC-like differentiated adipose-derived stem cells (dASCs) from rats were trypsinised and 80 × 10^6^ cells/mL were suspended in 25 μL diluted fibrinogen solution. The PHB strips were coated with 25 μL diluted thrombin (5 IU/mL) solution for 10 min and then the cell solution was added.	Functional gait test EMG, morphometry	The PHB strip seeded with cells provides a beneficial environment for nerve regeneration.
Ozer et al., 2018 [34]	Sciatic (mixed)/10 mm/8 weeks	Rats (30)	Poly-3-hydroxybutyrate (PHB) (molecular weight 454 kDa)	PHB (5 wt%) in chloroform by electrospinning method	Chitosan-coated PHB conduits were seeded with mesenchymal stem cells harvested from human iliac bone marrow (hMSC-bm)	Functional gait test, EMG, histology	PHB/chitosan-hMSC-bm nerve conduits may be a useful artificial guide for nerve regeneration.

**Table 2 biology-11-00706-t002:** Quality analysis of animal studies using ARRIVE guidelines [41] adapted for nerve regeneration treated with PHB scaffolds.

Study	Ethics	Control	Control 2	PHB Type	PHB Origin	Scaffold Fabric. Method	Nerve Gap Size	Nerve Studied	Period Evaluated	Surgical Procedure	Euthanasia Method	Species	Sex/Weight	Group Size & Distribution	Group Size Just.	Statistics	Complete Results	Precision Measures	Limitations	Conclusion > Objectives
Borkenhagen et al., 1998 [13]				●	●	●	●	●	●	●		●		●		●	●	●	●	
Ljungberg et al., 1999 [14]	●	●			●	●	●	●	●	●	●	●	●	●		●	●	●	●	●
Hazari et al., 1999a [15]	●	●		●	●	●	●	●	●	●		●	●	●		●	●	●		
Hazari et al., 1999b [16]	●	●		●	●	●	●	●	●	●		●					●	●		●
Young et al., 2002 [17]	●	●		●		●	●	●	●	●		●		●		●	●	●		●
Mohanna et al., 2003 [18]	●			●	●	●	●	●	●	●	●	●	●	●		●	●	●	●	●
Hart et al., 2003 [19]	●	●		●	●	●	●	●	●	●		●		●		●	●	●		●
Birchall et al., 2004 [20]	●	●			●	●	●	●	●	●	●	●	●	●			●		●	
Mohanna et al., 2005 [21]	●			●	●	●	●	●	●	●	●	●	●				●	●		●
Kalbermatten et al., 2008a [22]	●	●		●	●	●	●	●	●	●		●		●		●	●	●		
Kalbermatten et al., 2008b [23]	●			●	●	●	●	●	●	●		●		●		●	●	●		●
Kalbermatten et al., 2008c [24]	●			●	●	●	●	●	●	●		●		●		●	●	●		
Bian et al., 2009 [25]		●	●	●		●	●	●	●	●		●	●	●		●	●	●		●
Durgam et al., 2010 [27]		●	●	●	●	●	●	●	●			●	●	●			●	●		
Schaakxs et al., 2017 [33]	●	●		●	●	●	●	●	●	●	●	●		●		●	●	●		●
Ozer et al., 2018 [34]	●	●	●	●	●	●	●	●	●		●	●	●	●		●	●	●		●

● = Ethics: Declares to follow ethical guidelines for animal experimentation or approval by a Scientific Ethical Committee; Control: Experimental protocol has a control group; Control 2: Experimental protocol has 2 control groups; PHB type: declares the type of the PHB used; PHB origin: declares the origin of the PHB used; scaffold fabric method: explains the fabrication method for the scaffold; nerve gap size: declares the size of the nervous GAP; nerve studied: declares nerve studied; periods evaluated: declares the time in which the evaluations were carried out; surgical procedure: explains the surgical procedures performed; euthanasia method: explains the euthanasia method used; species: declares the animal species studied; sex/weight: declares sex and weight of animals at the beginning of the experimental protocol; group size and distribution: declares the size of the experimental group and explains the distribution of animals in the groups; group size just.: justifies the sample size; statistics: declares the statistical methods used for data analysis; complete results: presents complete results of the proposed the methodology; precision measures: reveals the precision values of the quantitative data (e.g., SD; SEM or IQ distance); limitations: states the limitations of the study; conclusion > objectives: conclusion consistent with the proposed objectives.

**Table 3 biology-11-00706-t003:** Analysis of the risk of bias in animal studies using the SYRCLE RoB tool [42] scale adapted for nerve damage and treated with PHB scaffolds.

STUDY	Selection Bias			Performance Bias		Detection Bias		Attrition Bias	Reporting Bias	Other
	Sequence	Baseline	Allocation	Random Housing	Housing Blinding	Random Outcome Asses.	Outcome Asses. Blinding	Incomplete Outcome Addr.	Sel. Outcome Rep.	Free of Other Problems
Borkenhagen et al., 1998 [13]	U	U	U	U	U	U	U	U	Y	N
Ljungberg et al., 1999 [14]	U	Y	N	U	U	U	U	Y	Y	N
Hazari et al., 1999a [15]	U	Y	U	U	U	U	U	U	U	N
Hazari et al., 1999b [16]	U	U	U	U	U	U	U	U	U	N
Young et al., 2002 [17]	U	U	U	U	U	U	U	U	U	Y
Mohanna et al., 2003 [18]	U	Y	U	U	U	U	U	U*	Y	Y
Hart et al., 2003 [19]	U	U	U	U	U	U	U	U*	U	N
Birchall et al., 2004 [20]	U	Y	U	U	U	U	U	U*	Y	Y
Mohanna et al., 2005 [21]	U	Y	U	U	U	U	U	U*	Y	N
Kalbermatten et al., 2008a [22]	U	U	U	U	U	U	U	U	U	N
Kalbermatten et al., 2008b [23]	U	U	U	U	U	U	U	Y	Y	Y
Kalbermatten et al., 2008c [24]	U	U	U	U	U	U	U	Y	U	Y
Bian et al., 2009 [25]	U	U	U	U	U	U	U	U	U	Y
Durgam et al., 2010 [27]	U	Y	U	U	U	U	U	U	Y	Y
Schaakxs et al., 2017 [33]	U	U	U	U	U	U	U	U*	Y	Y
Ozer et al., 2018 [34]	U*	Y	U	U	U	U	U	Y	Y	Y
	Y = There are explanations of the assignment	Y = reports sex, weight and species	Y = There are explanations of allocation concealment	Y = There are explanations of how the accommodation was hidden	Y = There are explanations of blinding of caregivers and/or researchers	Y = There are explanations of blinded analysis of animals	Y = Informs blinded evaluation of results	Y = Animal losses explained or states no animal losses.	Y = Reports positive and negative results	Y = Nothing unsual (bias)
	U = No explanation of the assignment. U* = mentions “randomly allocated”	U = Lack of baseline data	U = No explanation of allocation concealment	U = No explanation of accommodation concealment	U = No explanations of blinding of caregivers and/or researchers	U = No explanations of blinded analysis of animals	U = No information from blinded evaluation of the results.	U = Animal losses not declared. U* = Declared postoperative condition.	U = Does not report negative results	

## Data Availability

Data sharing is not applicable (Systematic Review of Literature).

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
