# Peer review of "Polyhydroxybutyrate (PHB) Scaffolds for Peripheral Nerve Regeneration: A Systematic Review of Animal Models"

_biology, 2022, doi:10.3390/biology11050706_

Round 1

Reviewer 1 Report

In the review paper “Polyhydroxybutyrate (PHB) scaffolds for peripheral nerve regeneration. A systematic review on animal models”, the authors reported a systematic literature review on the use of PHB in peripheral nerve regeneration.

I think that the topic is very interesting because it is very current and researchers are actually studying new biomaterials to be used as an alternative to the autograft. However, I have some concerns about the paper:

- Authors declare that the search was performed between September 2019 and March 2020. We are now in April 2022. Does it mean that you did not consider literature of the last 2 years? I think you should add papers (if any) published in these 2 years too.

- This review seems more a review about the analysis of the reproducibility/standardization as well as good quality of research papers than a review on the use of PHB in nerve regeneration. Of course, this is a crucial point that must be discussed (as you extensively did), but you gave very little attention to the results obtained in the different studies related to PHB. You should add more information about results, maybe adding a paragraph (the table is not enough) where you discuss about this point. Rather, I would delete the last column of the table “conclusion and main outcome” because is not complete and I would add more information on the text, in a dedicate paragraph.

- Also, there are few information about “additives”: concentration of factors? Methods of delivery? Number of cells used? Method of cell incorporation? These are all important information that are not available in the text.

- Beside PHB, there are others bio-polymers produced naturally by bacterial fermentation, such as PHA, that has been used as well in peripheral nerve regeneration. To give a broader view of the use of bio-polymers produced naturally by bacterial fermentation, you could also add this point, at least in the discussion.

- Also, you only considered in vivo studies on animal models and you also mention clinical studies. What about in vitro studies? This is also an important point that could be addressed/discussed.

Author Response

Thank you so much for reviewing our work. Your suggestions have been highly helpful to improve our manuscript. You may find our reply (R) below each comment (C).

In the review paper “Polyhydroxybutyrate (PHB) scaffolds for peripheral nerve regeneration. A systematic review on animal models”, the authors reported a systematic literature review on the use of PHB in peripheral nerve regeneration.

I think that the topic is very interesting because it is very current and researchers are actually studying new biomaterials to be used as an alternative to the autograft. However, I have some concerns about the paper:

C1: Authors declare that the search was performed between September 2019 and March 2020. We are now in April 2022. Does it mean that you did not consider literature of the last 2 years? I think you should add papers (if any) published in these 2 years too.

R1: Thank you for the observation. We conducted a new search and have not found any new articles since March 2020. Therefore, we only modified the dates of the search in the manuscript: “The search was performed between September 2019 and April 2022.”

C2: This review seems more a review about the analysis of the reproducibility/standardization as well as good quality of research papers than a review on the use of PHB in nerve regeneration. Of course, this is a crucial point that must be discussed (as you extensively did), but you gave very little attention to the results obtained in the different studies related to PHB. You should add more information about results, maybe adding a paragraph (the table is not enough) where you discuss about this point. Rather, I would delete the last column of the table “conclusion and main outcome” because is not complete and I would add more information on the text, in a dedicate paragraph.

R2: Thank you for your comment. We included a paragraph in the results section that carried out a brief analysis of the main results obtained in the selected studies. This information was also considered in the study discussion, which was rewritten based on this new information. However, we decided to keep the “main outcome” column in Table 1, as we understand this kind of data extracted from the selected studies is crucial for the understanding of the brief final message of each study.

C3: Also, there are few information about “additives”: concentration of factors? Methods of delivery? Number of cells used? Method of cell incorporation? These are all important information that are not available in the text.

R3: Thank you for the observation. We added more information on additives to the table.

C4: Beside PHB, there are others bio-polymers produced naturally by bacterial fermentation, such as PHA, that has been used as well in peripheral nerve regeneration. To give a broader view of the use of bio-polymers produced naturally by bacterial fermentation, you could also add this point, at least in the discussion.

R4:  Thank you for the comment. Polyhydroxybutyrate (PHB) is a Polyhydroxyalkanoate (PHA). PHAs are polyesters produced by numerous microorganisms. We added more information about PHB and other biopolymers produced naturally by microorganisms in the introduction.

C5: Also, you only considered in vivo studies on animal models and you also mention clinical studies. What about in vitro studies? This is also an important point that could be addressed/discussed.

R5: Thank you for the observation. In vitro studies on biomaterials for nerve regeneration, such as PHB, are numerous and of fundamental importance for understanding cellular mechanisms and allow the determination of possible candidate materials to obtain better regenerative results in this type of injury.  However, there is high heterogeneity in this type of study in addition to the lack of instruments that make it possible to analyze the quality and bias of these studies in a systematic way. In addition, we understand that studies in animal models are a step ahead regarding translationality compared to in vitro studies, so we decided to focus on these studies. However, we accept your suggestion to mention the importance of in vitro studies citing some more recent studies that analyzed the use of PHB for peripheral nerve regeneration.

Reviewer 2 Report

This is a well-organized and well-illustrated review article, with an important clinical message on the use of Polyhydroxybutyrate (PHB) scaffolds for peripheral nerve regeneration and should be of great interest to the readers. The review focused on the recent developments in the application of PHB scaffolds for the peripheral nerve injuries using animal models. Paragraphing is concise and good, and the article consists of major recent advancements in the field of scaffold usage for peripheral nerve regeneration. This review article deserves publication after some revisions listed below.

  1. Please check for typographical errors: eg; Line 18 va-riety, Line 33-Ma-ximize.
  2. The introduction section describes about the benefits of PHB scaffolds. However, I suggest the authors to add a brief section about the advantages of PHB over other natural scaffold materials like chitosan and briefly mention about the challenges of using natural scaffolds for nerve regeneration in the introduction part in brief.
  3. I suggest the authors to write a brief section about the limitations of currently available scaffold-materials for clinical translation and what measured must be followed for their successful clinical translation? Is PHB FDA approved? The authors reported that there is only one clinical study using PHB. What makes it difficult or challenging to use it in humans?

  1. The questions that author missed to address in conclusion:

There is always a dilemma on how to conclude a review article. Since the authors have deliberately summarized huge amounts of published results, it will go a long way. It would be helpful if they can provide their own thoughts that would in turn help in finding the areas that need to be addressed. For example, what are the factors that one needs to consider while choosing an ideal scaffold for nerve regeneration, what are the required criteria to overcome the toxicity associated with each of these scaffolds and what are the steps required for the fast transition of these materials for industrial scale up. In general, what measures need to be taken for the effective clinical translation of nerve regenerating scaffolds? Though natural scaffolds like chitosan are studied for decades, why there are only a handful of FDA approved scaffold materials? What limitations are hindering their clinical translation and in what direction does the future research need to be, to make the clinical translation possible?

Author Response

Thank you so much for reviewing our work. Your suggestions have been highly helpful to improve our manuscript. You may find our reply (R) below each comment (C).

This is a well-organized and well-illustrated review article, with an important clinical message on the use of Polyhydroxybutyrate (PHB) scaffolds for peripheral nerve regeneration and should be of great interest to the readers. The review focused on the recent developments in the application of PHB scaffolds for the peripheral nerve injuries using animal models. Paragraphing is concise and good, and the article consists of major recent advancements in the field of scaffold usage for peripheral nerve regeneration. This review article deserves publication after some revisions listed below.

C1: Please check for typographical errors: eg; Line 18 va-riety, Line 33-Ma-ximize.

R1: Thank you for the observation. We have made the necessary corrections throughout the entire abstract.

C2: The introduction section describes about the benefits of PHB scaffolds. However, I suggest the authors to add a brief section about the advantages of PHB over other natural scaffold materials like chitosan and briefly mention about the challenges of using natural scaffolds for nerve regeneration in the introduction part in brief.

R2: Thank you for the suggestion. We did include the advantages of PHB and we have added the challenges of using scaffolds made of natural polymers for nerve regeneration.

C3: I suggest the authors to write a brief section about the limitations of currently available scaffold-materials for clinical translation and what measured must be followed for their successful clinical translation? Is PHB FDA approved? The authors reported that there is only one clinical study using PHB. What makes it difficult or challenging to use it in humans?

R3: Thank you for the suggestion. We now explain within the introduction that this kind of polymer became widely available at the beginning of the 1990s, which provided opportunities for their evaluation as medical biomaterials (Williams and Martin, 2002). At the beginning, they were not targeted as implantable biomaterials and were thus lacking the quality which can get approval of the Drug Administrators (Ray and Kalia, 2017). PHB have caused prolonged and acute inflammatory responses, so the need was to produce PHB of high purity, check their biodegradation in vivo, fabrication of scaffolds and modify their surface (Ray and Kalia, 2017). The first film made of PHB for surgical applications was approved by the FDA in 2007 (Riaz et al., 2018).

Ray, S., & Kalia, V. C. (2017). Biomedical applications of polyhydroxyalkanoates. Indian journal of microbiology, 57(3), 261-269.

Riaz, S., Raza, Z. A., Majeed, M. I., & Jan, T. (2018). Synthesis of zinc sulfide nanoparticles and their incorporation into poly (hydroxybutyrate) matrix in the formation of a novel nanocomposite. Materials Research Express, 5(5), 055027.

Williams, S. F., & Martin, D. P. (2002). Applications of PHAs in medicine and pharmacy. Biopolymers, 4, 91-127.

C4: The questions that author missed to address in conclusion:

There is always a dilemma on how to conclude a review article. Since the authors have deliberately summarized huge amounts of published results, it will go a long way. It would be helpful if they can provide their own thoughts that would in turn help in finding the areas that need to be addressed. For example, what are the factors that one needs to consider while choosing an ideal scaffold for nerve regeneration, what are the required criteria to overcome the toxicity associated with each of these scaffolds and what are the steps required for the fast transition of these materials for industrial scale up. In general, what measures need to be taken for the effective clinical translation of nerve regenerating scaffolds? Though natural scaffolds like chitosan are studied for decades, why there are only a handful of FDA approved scaffold materials? What limitations are hindering their clinical translation and in what direction does the future research need to be, to make the clinical translation possible?

R4: Thank you for the suggestion. We added a final paragraph to the conclusion providing our own thoughts on the areas that need to be addressed.

Reviewer 3 Report

The authors reviewed an important field. However, the reviewer has the following concerns/comments:

  1. The type of article should be Mini-Review.
  2. The manuscript should be carefully proofread.
  3. Avoid single-sentence paragraphs.
  4. This review should be updated to include the articles published after March 2020.
  5. Moreover, most of the references were published before 2015. Are there fewer relevant publications with time?
  6. Title: “.” should be “:”.
  7. Table 2: one column regarding PHB characterization should be included. Besides origin, PHB’s molecular mass, purity, and other aspects are also critical.
  8. The review should also include the aspects of the research directions on PHB scaffolds for peripheral nerve regeneration.

Author Response

Thank you so much for reviewing our work. Your suggestions have been highly helpful to improve our manuscript. You may find our reply (R) below each comment (C).

The authors reviewed an important field. However, the reviewer has the following concerns/comments:

C1: The type of article should be Mini-Review.

R1: Thank you for the suggestion. We consider our manuscript to be a systematic review because we carefully followed the methodology and included all the resulting articles.

C2: The manuscript should be carefully proofread.

R2: Thank you for the observation. We carefully proofread the manuscript.

C3: Avoid single-sentence paragraphs.

R3: Thank you for the suggestion. You may no longer find single-sentence paragraphs within the manuscript.

C4: This review should be updated to include the articles published after March 2020.

R4: Thank you for the observation. We did a new search and have not found any new articles since March 2020. Therefore, we only modified the dates of the search within our manuscript: “The search was performed between September 2019 and April 2022.”

C5: Moreover, most of the references were published before 2015. Are there fewer relevant publications with time?

R5: Thank you for the observation. We found that there are fewer relevant publications on animal models with time. These kinds of polymer became widely available at the beginning of the 1990s, which provided opportunities for their evaluation as medical biomaterials (Williams and Martin, 2002). Most of the articles we found were published between 1995 and 2010. Then, the number of articles starts to drop. We found only 2 articles from the last 10 years. We included this information to the manuscript.

Williams, S. F., & Martin, D. P. (2002). Applications of PHAs in medicine and pharmacy. Biopolymers, 4, 91-127.

C6: Title: “.” should be “:”.

R6: Thank you for the observation. We changed the title to: “Polyhydroxybutyrate (PHB) scaffolds for peripheral nerve regeneration: a systematic review on animal models.”

C7: Table 2: one column regarding PHB characterization should be included. Besides origin, PHB’s molecular mass, purity, and other aspects are also critical.

R7: Thank you for the suggestion. We added the molecular mass of the polymers to the table. Twelve of the 16 articles reviewed used a PHB sheet commercialized by Astra Tech AB (Sweden) to make the nerve conduits but they do not give further details on the characteristics of the material. We are aware that the characteristics of the polymer are critical to treatment success. This is something that we now mention within the discussion.

C8: The review should also include the aspects of the research directions on PHB scaffolds for peripheral nerve regeneration.

R8: Thank you for the suggestion. We added a final paragraph providing our own thoughts regarding the areas that we consider to need to be addressed in future studies.

Reviewer 4 Report

The authors reports a litterature analysis about Polyhydroxybutyrate (PHB) scaffolds for peripheral nerve regeneration. Specifically, the authors focused on animal models studies. The review is sparce of results, collecting only tables. The data selction is well delineate. 

I suggest to publish the paper after a minor revision step, including the following points:

1) Introduction (line 41). Authors shoud expain the terms, to help a no-expert reader.

2) Intruduction (line 54). To enlarge the general scenario, I suggest to resport other cases of matrices based on different building-blocks, including peptides (e.g. 10.1002/chem.202102007), glyco-derivates (doi.org/10.1002/chem.202102950) other synthetic materials like polyrotaxanes (doi.org/10.1002/chem.201904446).

3) To improve the impact, author need to insert some appealing Figures about the studies they reported.

4) Include the chemical structure of the polymer and of some of its derivates. 

5) Results (line 161). The author need to report this information clearly.  

Author Response

Thank you so much for reviewing our work. Your suggestions have been highly helpful to improve our manuscript. You may find our reply (R) below each comment (C).

The authors reports a litterature analysis about Polyhydroxybutyrate (PHB) scaffolds for peripheral nerve regeneration. Specifically, the authors focused on animal models studies. The review is sparce of results, collecting only tables. The data selction is well delineate.

I suggest to publish the paper after a minor revision step, including the following points:

C1: Introduction (line 41). Authors shoud expain the terms, to help a no-expert reader.

R1: Thank you for the suggestion. You may now find an explanation to the terms: “The most commonly used techniques consist of directly suturing the ends of the cut nerve or closing the gap with an autograft—i.e., placing a piece of nerve harvested from the same individual— or an allograft —i.e., a piece of nerve harvested from another individual of the same species [2,3].”

C2: Intruduction (line 54). To enlarge the general scenario, I suggest to resport other cases of matrices based on different building-blocks, including peptides (e.g. 10.1002/chem.202102007), glyco-derivates (doi.org/10.1002/chem.202102950) other synthetic materials like polyrotaxanes (doi.org/10.1002/chem.201904446).

R2: Thank you for the comment. We included the suggested articles in the introduction.

C3: To improve the impact, author need to insert some appealing Figures about the studies they reported.

R3: Thank you for the suggestion. We added figures hoping to make the manuscript more appealing.

C4: Include the chemical structure of the polymer and of some of its derivates.

R4: Thank you for the suggestion. We included the chemical structure of the polymer and of some of its derivates within the introduction section.

C5: Results (line 161). The author need to report this information clearly. 

R5: Thank you for the observation. We rewrite the paragraph in order to make it clearer for the reader.

Round 2

Reviewer 1 Report

The authors answered all the points I requested and in my opinion the manuscript has improved. Therefore, the article can be accepted for publication

Reviewer 3 Report

none